# Aligning an optical interferometer with beam divergence control and continuous action space

**Stepan Makarenko**[1,2,*], **Dmitry Sorokin**[1], **Alexander Ulanov**[1], and **A. I. Lvovsky**[1,3]

[1]Russian Quantum Center, Moscow, Russia
[2]Moscow Institute of Physics and Technology, Russia
[3]University of Oxford, United Kingdom
[*]makarenko.sd@phystech.edu

**Abstract:** Reinforcement learning is finding its way to real-world problem application, transferring from simulated environments to physical setups. In this work, we implement vision-based alignment of an optical Mach-Zehnder interferometer with a confocal telescope in one arm, which controls the diameter and divergence of the corresponding beam. We use a continuous action space; exponential scaling enables us to handle actions within a range of over two orders of magnitude. Our agent trains only in a simulated environment with domain randomizations. In an experimental evaluation, the agent significantly outperforms an existing solution and a human expert.

**Keywords:** sim-to-real, robotics, optical interferometer

## 1 Introduction

Reinforcement learning (RL) demonstrates incredible success in simulated environments, surpassing a human expert in Atari [1], chess, shogi and Go [2], as well as in more complicated games such as Dota2 [3] and StarCraft [4]. In robotics, reinforcement learning shows remarkable results in pushing [5], grasping [6] and stacking objects [7]. RL enables robots to walk, overcoming different obstacles [8] and learn agile animal skills [9]. However, RL agents are not yet widely used in real robotics and can hardly be compared to humans in a physical environment.

The principal problems of real-world applications of RL are nonstationarity and stochasticity of the physical environment, complexity and time intensiveness of data acquisition as well as unsafety of training and evaluation of policies on real robots [10]. A common method to handle these challenges is to train an agent in a high-fidelity simulation (source domain) of the real environment (target domain). An agent can then be transferred to a physical robot. The main limitation of this approach is inevitable discrepancy between the source and target domains that lead to performance loss after transfer. This limitation can be mitigated with the randomization of simulator parameters [5, 9, 11].

A promising application domain for smart robotics is experimental optics. Optical physicists work with installations consisting of hundreds or thousands of elements. These setups need to be thoroughly aligned before measurements can be conducted. Dependent on the setup size, such a setup can require several hours of work by a group of experts on a daily basis. The automation of this procedure could drastically enhance the productivity of research groups. Since the alignment is an iterative decision-making process with a well-defined reward function, it is natural to formulate and solve it in terms of reinforcement learning.

The alignment of a large optical setup can typically be separated into modular tasks which can be completed in sequence. A common modular task is the alignment of a Mach-Zehnder interferometer (MZI), which consists in matching the optical modes of the electromagnetic waves in two paths so that they exhibit a visibility of unity when interfering with each other. The robotization of MZI alignment was studied by Sorokin *et al.* [12], where this problem was treated as a partially observable Markov decision process, with the observation being a sequence of interference patterns observed

on a camera as the relative phase of the two arms is being varied. They trained a discrete-action (VD D3QN [13]) agent, dubbed *Interferobot*, in a simulated environment with domain randomizations and successfully transferred the trained agent to a real setup. The trained agent achieved superhuman-level performance in terms of speed and quality of alignment.

However, the task of Ref. [12] was limited to matching the spatial positions and directions of the two otherwise identical beams. It is much simpler than those alignment tasks that occur in practice, which additionally require matching the geometric sizes of the beams, their divergence, and, in the case of pulsed lasers, arrival time. These additional features of the experiment result in more complex interference patterns (a richer observation space in RL parlance), hence requiring a more sophisticated agent for their interpretation.

**The main contribution of our work** is to develop a new agent for interferometer alignment, which addresses this challenge. Our solution is based on a different RL algorithm and features a number of important innovations. *First,* the action space was changed from discrete to continuous, which also changed the policy task from classification to regression. This enabled us to solve the issue arising in the discrete action space, consisting in the number of actions growing exponentially with the number of degrees of freedom [14]. However, it raises a new concern: the agent's actions towards the end of the alignment process need to be much finer than in the beginning; the action space spans over two orders of magnitude. We address this with our *second* innovation — exponential action rescaling, which allows the agent to effectively explore actions of different magnitudes. *Third,* by granting the agent a continuous action space, we run a risk to leave the safe space of experimental parameters, which may even result in damaging the equipment. We solve this issue by introducing penalties that discourage the policy from approaching boundaries. *Fourth,* we enhance the set of domain randomizations that help our agent perform better in the sim-to-real transfer. Our resulting agent[1] significantly outperforms the original Interferobot and a human expert in terms of both time and quality of the alignment.

## 2  Related works

RL agents for robotics can be trained either on real-world data or in simulation. The advantage of the former is that the agent receives hands-on experience of the environment in which it will be tested. The shortcoming is that the acquisition of a large dataset required for training is a complicated and time-consuming process. This is especially the case when the observations are vision-based. For example, the training of a grasping agent in Kalashnikov et al. [6] required several weeks, in spite of parallelization across 7 identical robots. This complication can be addressed with the help of behavior cloning: Vecerik et al. [15] trained an agent solving the insertion task using raw visual images using dozens of expert successful and failed trajectories. Hands-on training is simplified for agents without visual observations. For example, Haarnoja et al. [8] developed a quadruped walking robot that can generalize to unseen terrains with the training requiring as little as two hours. Another approach for training robot locomotion was presented by Yang et al. [16], who collected 4.5 minutes worth of data from a simple quadrupedal robot to model the robot's dynamics and used this model to real-time action planning.

An alternative approach is to train the agent in simulation and transfer it to a real-world system afterwards. To eliminate the discrepancy between the real and simulated environments, Tobin et al. [11] proposed a simple technique called domain randomization: they trained an object detection model on simulated images with different textures, lights, object and camera positions, and demonstrated that such a model achieves high accuracy in the real world. Peng et al. [5] trained a robotic hand to push an object using randomization of physical parameters such as friction, mass, damping, etc. Another example is the vision-based task of dexterous in-hand manipulation [17], which demonstrated the effectiveness of applying randomizations to both the physical parameters and image observations.

The automation of optical system alignment is constantly evolving. In 1987 Gabler et al. [18] demonstrated automated alignment of an optical fibre used an iterative algorithm that consequently finds the maximum of a photo-detector signal with respect to each of the three fibre movement axis. Fang and Savransky [19] considered automated alignment of a system of two lenses with 8 degrees of freedom. Acquiring a focal plane image, their algorithm performs principal component analysis and Kalman filtering to calculate proper control inputs.

---

[1] https://github.com/Stepan-Makarenko/RL_interferometer_alignment

Deep machine learning, particularly RL, has become increasingly popular in experimental optics. For example, Sun et al. [20] used an RL agent to stabilize a mode-locked laser by controlling waveplates and polarizers. RL algorithms are also routinely used to optimize optical communications [21], e.g. to route traffic in optical transport networks [22, 23].

# 3   Mach-Zehnder interferometer

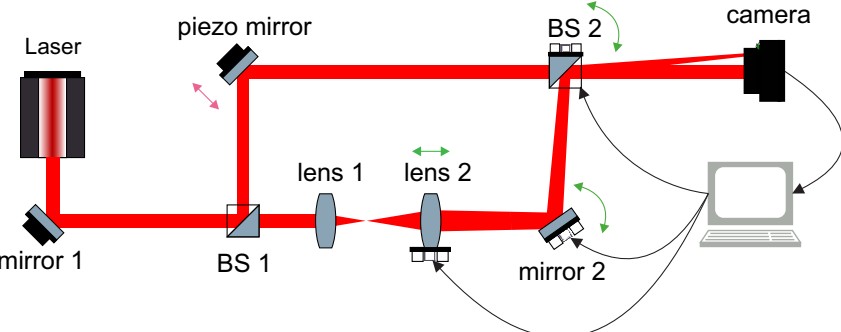

Figure 1: Conceptual scheme of the Mach-Zehnder interferometer. Lens 2, mirror 2 and BS 2 are motorized optical elements controlled by an RL agent.

Interference is a physical phenomenon that results from coherent addition of amplitudes of two or more overlapping waves; the resulting amplitude depends on the relative phase of the component waves. Interferometers, which are among the main instruments of experimental optics, use interference to precisely measure this phase difference.

In this paper we consider an MZI displayed in Fig. 1. A collimated (parallel) laser beam is divided by a beam splitter (BS 1). The two resulting beams propagate through different paths before being recombined, with the help of steering mirrors, by another beam splitter (BS 2) and viewed by a camera. One of the mirrors is mounted on a piezoelectric transducer to vary the relative phase $\varphi$ of the beams. One of the beam paths (the lower one in Fig. 1) contains a system of two lenses (telescope), which control the beam divergence and size. We use motorized mounts for mirror 2 and BS 2; lens 2 is mounted on a motorized translation stage. Aligning the interferometer requires precise matching of the two beams in terms of their transverse positions, directions, radii and divergences.

The images from the camera serve as observations for our RL agent. Examples of acquired images are shown in Fig. 2. The curved shape of the interference fringes is a consequence of different divergences of the two beams. We apply an asymmetric sawtooth voltage pattern to the piezo, akin to Ref. [12], to view the temporal dynamics of fringes. The amplitude of the piezo motion corresponds to a phase difference of about $2\pi$. For a misaligned interferometer, the piezo motion causes transverse displacement of the fringes visible in Fig. 2 (b-d), which permit the agent to extract the information about the sign of the difference of the beams' angles. When fully aligned, the interference will appear as a single blinking spot [Fig. 2 (a)].

The complexity of alignment arises because changing the angular orientations of the mirrors simultaneously affects the position and angle of the lower beam, while the lens movement changes the lower beam radius and divergence. Moreover, images received from the camera can be corrupted by noise, aberrations, ambient light and dust. The problem is additionally complicated by the inaccuracy of mounts and positioners whose relative action noise is about $4\%$.

The position-dependent intensity of the interference pattern can be written as $I(x, y, \varphi) = \frac{1}{2}|E_{\text{upper}}(x, y)e^{i\varphi} + E_{\text{lower}}(x, y)|^2$, where $E_{\text{upper/lower}}(x, y)$ is the amplitude of each beam. The quality metric of alignment, *visibility*, is defined as

$$V = \frac{\max_\varphi(I_{\text{tot}}(\varphi)) - \min_\varphi(I_{\text{tot}}(\varphi))}{\max_\varphi(I_{\text{tot}}(\varphi)) + \min_\varphi(I_{\text{tot}}(\varphi))} \tag{1}$$

where $I_{\text{tot}}(\varphi) = \iint I(x, y, \varphi)dxdy$ is a total light intensity in the image plane with the phase difference $\varphi$ between the arms. The intensity depends on time because of periodic piezo movement

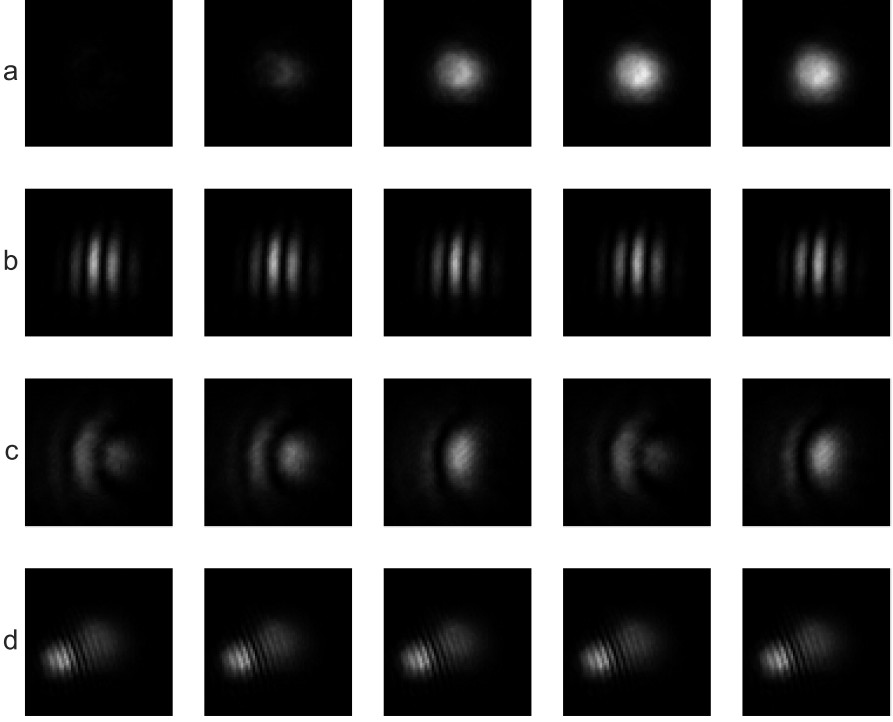

Figure 2: Camera images from the Mach-Zehnder interferometer. (b-d) Example images acquired with a misaligned interferometer. The images in each row correspond to varying path length differences of the interferometer arms.

that varies the phase difference between the two interferometer arms. The visibility lies in $[0, 1]$ range, where $V = 1$ corresponds to a perfectly aligned interferometer.

## 4 Background

We consider a standard partially observable Markov decision process (POMDP) setting where the purpose of an agent is to maximize its cumulative reward during the policy execution. The observation of the environment at timestep $t$ is defined as $o_t \in \mathcal{O}$ and is sampled from the distribution $o_t \sim U(o_t|s_t)$, where $s_t \in \mathcal{S}$ is the state. The policy of our agent is a deterministic function $a_t = \pi(o_t)$ yielding an action, which is an element of the action space $\mathcal{A}$. Following the action, the agent receives a reward $r_t : \mathcal{S} \times \mathcal{A} \to R$ and the next observation $o_{t+1} \sim U(o_{t+1}|s_{t+1})$, where $s_{t+1}$ is produced according to a latent transition distribution $s_{t+1} \sim F(s_{t+1}|s_t, a_t)$. The discounted reward sum (return) from timestep $t$ in each episode is $R_t = \sum_{i=t}^{T} \gamma^{i-t} r_i$, where $\gamma \in [0, 1]$ is the discount factor and $T$ is the horizon of the episode.

The agent's objective is to learn the policy $\pi^*$ that maximizes the expected return $J(\pi) = E_{\tau \sim p(\tau|\pi)}[R_0]$ where $p(\tau|\pi)$ is the distribution of trajectories $\tau = (o_0, a_0, o_1, a_1, ..., a_{T-1}, o_T)$ produced by policy $\pi$:

$$\pi^* = \arg\max_{\pi} J(\pi). \tag{2}$$

**Policy gradient.** The policy $\pi$ is defined by parameters $\theta$ (i.e. $\pi = \pi_\theta$), so the optimization (2) is performed with respect to the parameters: $\theta^* = \arg\max_\theta J(\pi_\theta)$. A common approach to such a problem is policy gradient [24], which iteratively improves the policy in terms of expected return via gradient ascent:

$$\theta_{t+1} = \theta_t + \alpha \frac{\partial J(\pi_\theta)}{\partial \theta}, \tag{3}$$

where $\alpha$ is the learning rate. The algorithms of this family vary by the method of approximating the unknown $J(\pi_\theta)$.

A common approach to continuous-action Markov decision processes is TD3 [25] which is an extension of another popular algorithm DDPG [26]. This algorithm uses three neural networks: one (actor) for a deterministic policy and two (critics) for evaluating the action-state values $Q(s_t, a_t)$. Using two critics allows the algorithm to suffer less from overestimating the Q-values; in addition, the algorithm smoothens the Q-functions by adding Gaussian noise to the target actions when updating the parameters of the critics. Gaussian noise is also added to each policy action during the training to encourage exploration.

**Domain randomization.** To lower the performance losses associated with the transferring of the agent from a simulated to real-world environment, domain randomization is used. The agent is trained for a set of tasks with different dynamics $\hat{F}(s_t|s_{t-1}, a_{t-1}, \mu_{\mathrm{sim}}) \approx F(s_t|s_{t-1}, a_{t-1}, \mu_{\mathrm{real}})$ and sensor noise models $\hat{U}(o_t|s_t, \mu_{\mathrm{sim}}) \approx U(o_t|s_t, \mu_{\mathrm{real}})$ where $\mu_{\mathrm{sim/real}}$ are sets of environment parameters. Whereas we do not know the actual $\mu_{\mathrm{real}}$ of the real environment, varying $\mu_{\mathrm{sim}}$ in a range containing $\mu_{\mathrm{real}}$ helps to improve generalization and transfer quality.

## 5 Our method

In this work, we apply continuous control reinforcement learning methods to align the MZI. As mentioned above, the alignment is performed step by step and the actions depend on the interference patterns observed by the agent, so it can be naturally viewed as a POMDP.

The agent is trained in simulation and evaluated on a real interferometer. The training algorithm is listed in Appendix A and the MZI simulator is described in Appendix B.

**State, observation and action.** The state of the environment is a vector consisting of the transverse position $(x, y)$, direction angle $(\alpha_x, \alpha_y)$, radius $r_{\mathrm{lower}}$ and divergence curvature $\rho$ of the lower beam in the plane of the camera. The position and transverse direction of the upper beam are assumed to be zero. This state is fully determined by the angles of the mirror and the beam splitter and the position of the lens. The range of allowed states, listed in Table 1, is restricted by the requirement that the beam remains visible on the camera and the position of the lens stays within the travel range of the translation stage.

The observation is a set of 16 consecutive $64 \times 64$ images acquired by the camera during one period of the piezo mirror. The actions are five-dimensional vectors specifying the relative angular deflections of both mirrors along the $x$ and $y$ axes and the linear displacement of the lens with respect to their current positions. Each action vector component lies in the $[-1, 1]$ range, where the values of $\pm 1$ correspond to the maximum and minimum values as listed in Table 1. The absolute values of each action are restricted to the interval $[2.5 \cdot 10^{-3}, 1]$, because smaller actions do not produce observable changes in the interference pattern and fall within the uncertainty range of motorized mounts.

**Episode and reset.**

To compare the performance of our agent with that of Sorokin et al. [12], we keep the episode length equal to 100 actions. At the beginning of each episode, the beam in the lower path is misaligned by setting the mirror angles and lens position to random values within the allowed range.

**Reward.** Under the normal alignment procedure, the reward at each step is positive and consists of two terms:

$$R = V - \log(1 - V). \tag{4}$$

The first term is the visibility, which, as discussed above, is the primary metric for the interferometer alignment quality. The second term rewards high quality of the final alignment, which is critical for optical experiments; it tends to infinity for $V \to 1$ [12].

| Element angle / position | mirror 2, x | mirror 2, y | BS 2, x | BS 2, y | lens 2 |
|---|---|---|---|---|---|
| Amplitude of max deflection | $2.6 \cdot 10^{-3}$ | $1.8 \cdot 10^{-3}$ | $1.3 \cdot 10^{-3}$ | $0.9 \cdot 10^{-3}$ | 7.5 |

Table 1: Maximum deflection of each optical element. Mirrors angles are given in radians, lens positions in millimeters.

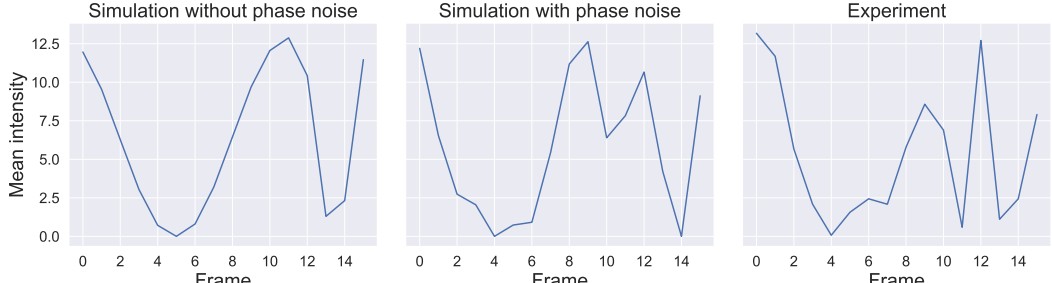

Figure 3: Effect of phase noise. The plots display the integrated intensity of the interference pattern as the path length difference is varied. The behavior becomes closer to real after applying phase noise.

However, if the agent proposes an action that takes one of the controls out of the boundaries defined in Table 1, the episode is terminated to avoid damage to the equipment. Additionally, the agent is penalized with the reward of $P = -0.04$. This penalty is important during early stages of the training, when the reward (4) is normally close to zero. A small negative reward for unsafe actions will train the agent to be aware of the bounds, but will not discourage exploration. The specific value of $P$ was handpicked via experimentation. On the other hand, when the agent is well-trained, it normally receives a significant positive reward for each step. At this stage, terminating an episode has a major negative effect on the return, thereby strongly discouraging unsafe actions.

**Domain randomization.** To learn a more generalized policy and facilitate sim-to-real transfer, we introduce several domain randomizations, which are used during the training. The only randomization that changes the dynamic of the environment $F(s_{t+1}|s_t, a_t)$ is the beam radius randomization within $\pm 20\%$ of the measured radius $r = 0.71$ mm. The randomization is applied at the beginning of each episode.

Additionally, we introduce the following randomizations of the observation $U(o_t|s_t)$. First, to address the noise of the camera detector, scattering, air fluctuations and dust effects, we add Gaussian noise to each pixel acquired by the camera. The standard deviation of the noise is 20% with respect to the simulated intensity value. Second, we rearrange cyclically the images within the video sequence to account for the randomness of the camera trigger. In addition, we randomize the duty cycle (fraction of time spent on the forward and backward passes) of the piezo. Third, we vary the beams' brightness by $\pm 30\%$ to model the variance of the camera exposure. All these randomizations are applied to each time step (i.e. they are constant for the 16 frames acquired in each step).

The randomizations described above have been used in Sorokin et al. [12] and have proven their effectiveness. An additional randomization we introduce in this work addresses irregularities in the motion of the piezo mirror and the frame rate, as well as fluctuations in the air density in different arms of the interferometer, all of which lead to random variation of the optical path length difference between the two MZI arms. To simulate these effects, we add Gaussian noise with the standard deviation of $0.5$ rad to this phase difference.

**Action rescaling.** As the alignment progresses, the agent's actions become smaller and more precise. The typical action magnitude decreases during an episode by about two orders (as illustrated in the experimental Section 6 below). It is therefore desirable that the exploration noise also decrease with the action magnitude. We satisfy this by setting the agent's neural network to output a "raw action" value $a_0 \in [-1, 1]$, from which the actual action is calculated according to

$$a = \begin{cases} \text{sign}(a_0) \cdot 1000^{|a_0|-1} & \text{if } |a_0| > 0.17 \\ 0 & \text{if } |a_0| \leq 0.17 \end{cases} \tag{5}$$

This transformation produces rescaled actions with absolute values $|a| \in \{0\} \cup [2.5 \cdot 10^{-3}, 1]$.

**Algorithm and network architecture.** We use the standard TD3 algorithm with handpicked hyperparameters to produce a deterministic policy, which yields the raw action $a_0$ as described above. For both critics and the actor, we use the VGG-16 [27] architecture, modified as follows. The number of convolutional layers in the encoder is set to 8, followed by three MLP layers, with no dropout. We have chosen this architecture because max pooling operations help reducing the

|  | $V \geq 0.92$ | $V \geq 0.95$ | $V \geq 0.98$ |
|---|---|---|---|
| Human | 93.9 (**0%**) | 103.6 (**0%**) | 129.6 (10%) |
| TD3 (our agent) | **56.16** (**0%**) | **75.06** (**0%**) | **120.1** (**4%**) |
| Interferobot | 98.7 (7.6%) | 116.1 (7.6%) | 156.4 (10.6%) |

Table 2: Comparison with a human expert and the original Interferobot. The time required to reach the visibility thresholds of 0.92, 0.95 and 0.98 are shown, together with the percentage of episodes the threshold has not been reached (in parentheses).

overfitting and sensitivity to individual pixel noise. We use orthogonal initialization in all models as we found it to ensure faster convergence.

We train the agent with a discount factor $\gamma = 0.8$; this relatively short reward sight inspires our agent to reach high visibility faster. A Gaussian exploration noise is added to the raw action $a_0$ with the standard deviation decreasing exponentially from 0.5 to 0.02 during the training. Although the variance of this noise is independent of the magnitude of $a_0$, its effect on the actual action $a$ is proportional to its magnitude due to the exponential dependence (5). The total number of steps is $10^6$ and the replay buffer size is $10^5$. Updates are performed every ten steps. The whole training on an NVidia RTX 2080 GPU takes about 26 hours.

## 6 Experimental evaluation

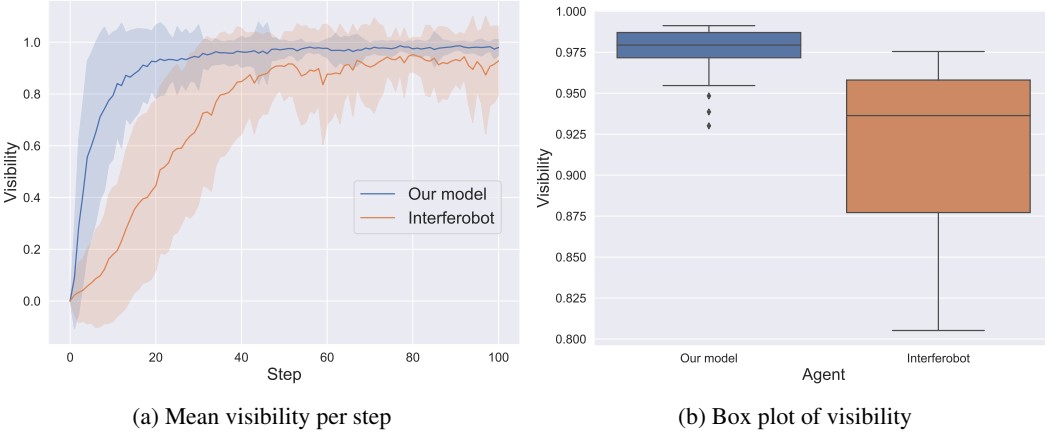

(a) Mean visibility per step

(b) Box plot of visibility

Figure 4: Comparison of Interferobot and our agent, demonstrating the advantage of the latter in convergence speed and the final alignment quality. As evident from the box plot (b), the 25-75 interquartile range for our agent is significantly narrower than that of Interferobot, indicating higher stability of the former. All results are averaged over 50 episodes.

For our experiments, we build an MZI according to the scheme shown in Fig. 1. We use a continuous HeNe laser with wavelength $\lambda = 632$ nm, Newport Picomotor mirror mounts, a Standa 8MT167-25LS linear translation stage, a CMOS camera with 16 fps acquisition rate, and a high-bandwidth photodetector to precisely measure the visibility. In addition to our agent, we also train and evaluate the original Interferobot [12] with the action space extended to include the lens movement. For each agent, we run 50 episodes of evaluation with 100 timesteps. Figure 4 shows the evaluation results. Our agent significantly outperforms Interferobot in terms of the alignment time, final visibility (averaged over the last 40 steps of each episode) and its variance.

Additionally, our agent has been compared with a human expert, who aligned the interferometer 10 times by manually turning the controls on the physical setup. As evidenced by Table 2, our agent surpasses the human both in the speed and quality of the alignment.

A common trick used by human experts in aligning optics is to turn two mirrors steering the same beam by equal and opposite angles. This action moves the position of the beam without changing

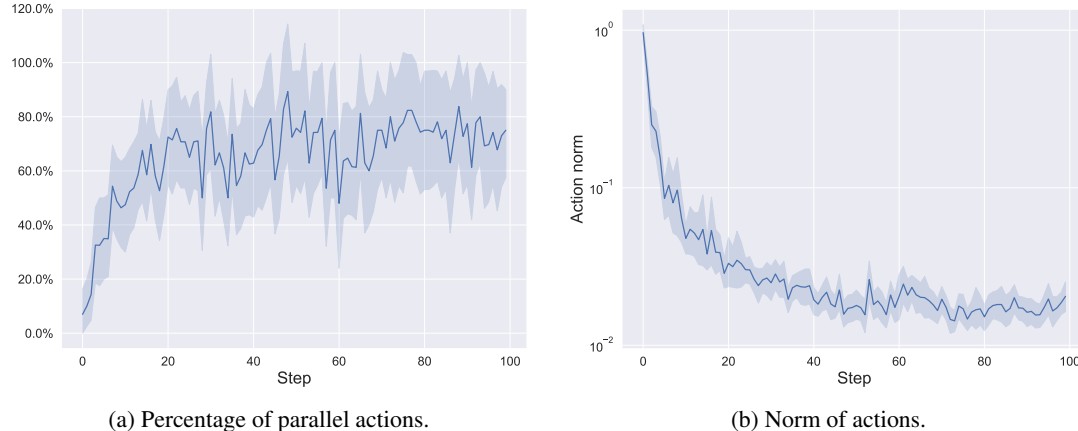

| (a) Percentage of parallel actions. | (b) Norm of actions. |

Figure 5: Interpreting the agent's policy. a) Percentage of actions that change the distance between beams while conserving the directions increases with the step number. b) Mean norm of actions decreases with the step number.

its direction. Fig. 5a shows that our agent learns this method and uses it extensively: by the end of the procedure, about 70% of the actions contain such parallel movements. Fig. 5b shows the action norm as a function of the step number. The descending trend means that at the beginning our agent uses large actions to obtain rough alignment and decreases the action size to fine-tune the interferometer.

Table 3 presents the results of an ablation study. A standard TD3 agent does not outperform the original Interferobot and achieves a mean visibility of $V = 0.83$. Action rescaling and phase noise randomizations significantly boost the performance of our agent, leading to a visibility of $V = 0.98$. It is notable that the phase noise randomization slightly decreases the mean visibility of the Interferobot but improves the standard deviation. This bias-variance trade-off can be due to the relative simplicity of the Interferobot model which cannot capture richer observations produced by this randomization.

| Model | mean visibility for last 40 step | standard deviation |
|---|---|---|
| TD3 + AR + PN | **0.98** | **0.03** |
| TD3 + AR | 0.95 | 0.06 |
| Interferobot | 0.92 | 0.12 |
| Interferobot + PN | 0.90 | 0.08 |
| TD3 | 0.83 | 0.18 |

Table 3: Comparative evaluation with ablated agents and the original Interferobot [12]. PN: phase noise; AR: action rescaling.

## 7    Conclusion

We demonstrated a novel RL algorithm for the automatic alignment of an optical interferometer, which contains a system of lenses in one of its optical paths. Such a setup results in richer observation and action spaces and constiututes a better approximation of a general problem of achieving mode matching between two arbitrary Gaussian beams. Our reinforcement learning agent with a continuous action space solves the alignment problem successfully, surpassing both a previous solution and a human expert. Important innovations include exponential scaling of the action space, an additional domain randomization that helps the agent generalize to a real setup after training in simulation as well as a discrete reward structure that trains the agent to avoid unsafe actions. Our solution will work for any Mach-Zehnder interferometer with $2 \times 2$ degrees of freedom (mirror angles) controlling the position and direction, and one additional degree of freedom (lens position) controlling the width and divergence of one of the beams.

## 8 Acknowledgments

We acknowledge support from Russian Science Foundation (19-71-10092).

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
