# OpenReview forum: "Aligning an optical interferometer with beam divergence control and continuous action space"
_robot-learning.org/CoRL/2021/Conference — CoRL2021 Poster_

### Official Review · Reviewer_ENy6 · 2021-07-21

**Originality:** Poor
**Technical Quality:** Good
**Clarity Of Presentation:** Very Good
**Impact:** 3

**Recommendation:**

Weak Reject: I recommend rejecting the paper, but will not argue for my recommendation if the majority of other reviewers have a different opinion.

**Summary:**

The paper presents an application of Deep RL to the problem of aligning an optical interferometer. In simulation, TD3 is used to learn a controller that is able to manipulate different parts of the interferometer such as mirror orientations and lens locations. Domain randomization is applied for succesful transfer from simulation to the actual physical system. The resulting policy is able to control the interferometer parts more precisely and quicker than a human expert and another deep RL baseline algorithm. Since the action space on the actual system can span several orders of magnitude, the method applies an exponential scaling to the actions after sampling actions from the interval [-1, 1].

**Issues:**

- Although I'm quite sure the paper uses the same simulator as in [12], it is never actually introduced


**Reviewer Expertise:**

Good: General knowledge of the area

**Strengths And Weaknesses:**

Strengths:
- Application of a deep RL algorithm to a real world problem
- It's interesting to see that the learned policy applies techniques also developed by human experts to solve the task (parallel actions)

Weaknesses:
- Very incremental

**Summary Of Recommendation:**

The work is heavily inspired by [12] in terms of the problem to be solved and the methodology applied. The main differences seem to be the use of TD3 instead of double dueling DQN, action space scaling to handle the multiple orders of magnitude, and an additional randomized parameter of the domain. In my opinion, this is too incremental to recommend it for acceptance.

---

> ### Author Response · Authors · 2021-08-30
> **Response to Reviewer ENy6**
>
> Thank you for the thorough reading of our work and insightful comments.
> > The work is heavily inspired by [12] in terms of the problem to be solved and the methodology applied.
>
> Please see our response to Reviewer rhsP above.
>
> > Although I'm quite sure the paper uses the same simulator as in [12], it is never actually introduced
>
> We added a description of the simulator in the Appendix of the new version.

---

> > ### Comment · Reviewer_ENy6 · 2021-09-06
> > **Response to Author Rebuttal**
> >
> > This comment is to acknowledge that I have read the authors' revisions and responses to the reviewers.
> >
> > I appreciate the authors' effort in addressing the concerns. While I understand that the problem solved is harder than the original Interferobot from an engineering point of view, I don't see any new challenges from a theory/algorithmic point of view. To the algorithm it's just numbers. I do like the way the action space over several orders of magnitude is handled and, given the fact that the problem is solved on a real setup, I am willing to increase my score to weak reject and would also be fine if the paper is accepted.

---

### Official Review · Reviewer_rhsP · 2021-07-22

**Originality:** Good
**Technical Quality:** Very Good
**Clarity Of Presentation:** Excellent
**Impact:** 3

**Recommendation:**

Strong Accept: I recommend accepting the paper and will argue for my recommendation even if other reviewers hold a different opinion.

**Summary:**

The goal of this paper is to use reinforcement learning to solve a practical problem in optics. In particulars, the authors use RL to align an optical interferometer. The problem seems perfectly suited for RL, as there is a clear observation space, action space and reward function, and the problem can be simulated with relative ease. The authors follow a sim-to-real approach, training their agent entirely in simulation, and then transferring it to a real world system. To ensure effective sim-to-real transfer, the authors make use of domain randomization, in which they both randomize the dynamics (a little), and the observations (a lot).

The closest prior work is that of Interferobot [1], on which this work closely builds. The primary improvement over Interferobot is the use of a continuous action space and an exponential action scaling scheme that allows the agent to take actions across two orders of magnitude. The authors use TD3 as their RL algorithm. The paper re-uses some domain randomizations from prior work, and also introduces some new ones.

[1] D. Sorokin, A. Ulanov, E. Sazhina, and A. Lvovsky. Interferobot: aligning an optical interferometer by a reinforcement learning agent. In Advances in Neural Information Processing Systems.

**Issues:**

I'll leave some typos here:

Typos
Line 91: “becomes” → “has become”
Line 105: “mororized” → “motorized”
Line 107: “required” → “requires”


**Reviewer Expertise:**

Good: General knowledge of the area

**Strengths And Weaknesses:**

Strengths
- Practical solution to a real world problem. This is not something that you can say for most papers submitted to CoRL (as most robotics is limited to toy-settings, not real world ones).
- Clear presentation
- Outperforms prior work by a decent margin

Weaknesses
- Somewhat incremental taking into account the prior work (Interferobot)

**Summary Of Recommendation:**

The paper is a bit incremental taking into account prior work (as I mentioned above), which is why I am keeping my recommendation at a weak accept.

After author response: I found the author response to be quite compelling, so I am upgrading my rating to a strong accept. While I am not an optics expert, the works appears to be tackling a much harder problem than the previous work, and while each of the change might be incremental by itself, there are quite a few such changes that come together to solve a real world problem, which makes this paper a strong candidate for acceptance to CoRL (in my view).

---

> ### Author Response · Authors · 2021-08-30
> **Response to Reviewer rhsP**
>
> Thank you for the thorough reading of our work and insightful comments.
>
> >The paper is a bit incremental taking into account prior work.
>
> Compared to the original Interferobot paper, we introduce an optical scheme with an important additional element - the lens, whose physical properties are different from that of mirrors and which significantly complicate the process of interferometer alignment. In the previous setting, the beams had identical sizes and divergences, and the four degrees of freedom only affected one of the beams’ position and direction. This results in relatively simple interference patterns with straight fringes. In principle, the alignment under such conditions could be achieved simply by computing the Fourier transform of the interference pattern from which the angle between the two beams could be directly deducted.
>
> The situation becomes significantly different with the introduction of the lens. Now the radius and the wavefront shape of the beam are variable, so the interference patterns are more complex; one can see the difference by comparing Fig. 2 in our paper and Fig. 2 in the Interferobot paper. These new patterns can no longer be interpreted using simple mathematical tools. In the language of RL, the addition of the lens results in a much richer observation space and requires a more complicated policy. It also greatly increases the practical value of the agent: in a real optical setting, one needs to match two beams of arbitrary sizes and divergences, which is a nontrivial task even for a trained human professional.
>
> As the problem at hand is significantly more challenging, we had to introduce major changes to the RL solution - in fact, write an entirely new agent from scratch and introduce several important innovations.  First, the action space was changed from discrete to continuous, which also changed the policy task from classification to regression. This required us to implement a different RL algorithm. Second, working with continuous actions that span two orders of magnitude raise, to our knowledge, a new previously unconsidered challenge, which we address with exponential action rescaling, and we believe that this approach may be suitable for other robotic tasks like dexterous manipulation. Third, by granting the agent the possibility to freely manipulate every optical element with a large amplitude, we run a risk to damage the equipment. We solve this issue by introducing penalties that discourage the policy from approaching boundaries. Fourth, we have introduced a new randomization that handles important physical effects of air fluctuations and also allows us to vary a piezo generator frequency in a small range. Finally, we address many shortcomings of the original Interferobot paper: poor scaling, handpicking discrete actions, inability to manipulate more than one element at each step of execution.
>
> In summary, our work is in fact drastically different from the original paper in terms of both the problem being solved and the solution implemented “under the hood” of the agent. We added a discussion of these differences in the new version of the manuscript.
>
> We are grateful for the Reviewer’s help with the typos, they are fixed in the new version.

---

> ### Comment · Reviewer_rhsP · 2021-09-02
> **Upgrading to strong accept**
>
> I found the author response to be quite compelling, so I am upgrading my rating to a strong accept. While I am not an optics expert, the works appears to be tackling a much harder problem than the previous work, and while each of the change might be incremental by itself, there are quite a few such changes that come together to solve a real world problem, which makes this paper a strong candidate for acceptance to CoRL (in my view).

---

### Official Review · Reviewer_GxfU · 2021-07-24

**Originality:** Fair
**Technical Quality:** Very Good
**Clarity Of Presentation:** Good
**Impact:** 2

**Recommendation:**

Weak Accept: I recommend accepting the paper, but will not argue for my recommendation if the majority of other reviewers have a different opinion.

**Summary:**

This paper proposes using reinforcement learning to train a robot for aligning an interferometer. They extend setting to a richer observation and action spaces that are closer to the real-world. To work with continuous action spaces, TD3 algorithm is used. Additional novel components include predicting the actions in the log space to handle the large range of actions, and adding phase noise for domain randomization.

**Issues:**

- Clarify how action rescaling is related to exploration noise.
- Discuss how the method may / may not be used in other real-world applications.

**Reviewer Expertise:**

Fair: Some knowledge of the area

**Strengths And Weaknesses:**

Strengths:
- The experimental evaluation, particularly on the real system, is convincing, and shows that the proposed approach can outperform the baseline, and a human. Further, the method discovers parallel movement, which is interesting.

Weaknesses:
- The section on action rescaling (line 216) was a little confusing — specifically, it wasn't clear how this is related to the exploration noise. It seems like exploration noise is reduced by decaying the standard deviation of the Gaussian noise (line 230).
- It might be helpful to have a discussion of whether the proposed system is specific to the setup considered, or whether it could generalize to other related systems.

**Summary Of Recommendation:**

The paper extends prior work to a practical real-world system, and can potentially be used to implement reinforcement learning in other real-world systems.

---

> ### Author Response · Authors · 2021-08-30
> **Response to Reviewer GxfU**
>
> Thank you for the thorough reading of our work and insightful comments.
> > The section on action rescaling (line 216) was a little confusing — specifically, it wasn't clear how this is related to the exploration noise. It seems like exploration noise is reduced by decaying the standard deviation of the Gaussian noise (line 230).
>
> This is correct: the exploration noise is reduced in the course of the training. We added a sentence discussing the effect of the exploration noise on actual and raw actions (a and a0, respectively) Note that a full listing of our algorithm is provided in the Appendix.
>
> > It might be helpful to have a discussion of whether the proposed system is specific to the setup considered, or whether it could generalize to other related systems.
>
> Our solution will work for any Mach-Zehnder interferometer with 2⨉2 degrees of freedom (mirror angles) controlling the  position and direction of the beam, and one additional degree of freedom (lens position) controlling the beam width and divergence. We clarified this in the new version of the paper.

---

### Official Review · Reviewer_PgwD · 2021-07-24

**Originality:** Good
**Technical Quality:** Very Good
**Clarity Of Presentation:** Very Good
**Impact:** 2

**Recommendation:**

Weak Accept: I recommend accepting the paper, but will not argue for my recommendation if the majority of other reviewers have a different opinion.

**Summary:**

This paper is concerned with the problem of automatically aligning elements for optical setups. The proposed method is to train a policy to adjust the components positions based on the current interferometer output. The policy is trained in simulation and then deployed on a real setup. The paper's contributions are an extension to the prior simulated benchmark for this task and exponential action rescaling to perform coarse-to-fine adjustments.

**Issues:**

My concern with this paper is whether CoRL is an appropriate venue. The work is interesting and seems significant for the optics community, but I do not have the qualifications to judge whether the problem setting and results are realistic and useful. From an RL perspective, the method is sound but not particularly novel. I could see the action rescaling being applicable to other robotics problems (fine grained manipulation tasks) but this is a small part of the paper. Overall, I would prefer for this paper to be submitted to an optics venue.

**Reviewer Expertise:**

Fair: Some knowledge of the area

**Strengths And Weaknesses:**

The paper is clear in how it describes the problem setting and method. The experiments are well reported, and the human baseline shows the speed advantage of the RL agent. The introduction discusses installations that consist of "hundreds or thousands of elements", but the task presented only has 9. Will this method scale to larger settings? RL is notoriously bad at scaling to large action spaces without some sort of modularity or hierarchy.

**Summary Of Recommendation:**

While this is an interesting paper with potential impact for optics practitioners, CoRL seems like the wrong venue for this work.

------
Update after author response.
I agree the work is well done, and I am fine with it being accepted to CoRL.

---

> ### Author Response · Authors · 2021-08-30
> **Response to Reviewer PgwD**
>
> Thank you for the thorough reading of our work and insightful comments.
> > The introduction discusses installations that consist of "hundreds or thousands of elements", but the task presented only has 9. Will this method scale to larger settings?
>
> In spite of a typical tabletop setup consisting of multiple elements, it has a modular structure; individual modules have only a few elements and can be aligned in sequence. MZI (which represents a more general task of precise matching of two beams coming from different parts of the setup) is one of the most commonly occurring modules. We have further clarified this point in the new version of the paper.
>
> > My concern with this paper is whether CoRL is an appropriate venue. The work is interesting and seems significant for the optics community, but I do not have the qualifications to judge whether the problem setting and results are realistic and useful. From an RL perspective, the method is sound but not particularly novel. I could see the action rescaling being applicable to other robotics problems (fine grained manipulation tasks) but this is a small part of the paper. Overall, I would prefer for this paper to be submitted to an optics venue, as discussed in our response to Reviewer rhsP below
>
> Our work is within the scope of CoRL as defined on the conference web page (https://www.robot-learning.org/author-information/call-for-papers), with a particularly significant overlap existing in the topics of “Reinforcement learning” and “Applications of robotic learning”
> Our work is interdisciplinary, and we believe it to be of interest to both communities. Specifically for the robotics community, the value is in the new application of robotic learning that has not been previously explored. As Referee 3 puts it, our work presents a “Practical solution to a real world problem. This is not something that you can say for most papers submitted to CoRL (as most robotics is limited to toy-settings, not real world ones).” While this application has been discovered in the first Interferobot paper, the present manuscript solves a more complex task and uses more advanced RL techniques to achieve successful robotic learning in a setting of much higher practical relevance, as discussed in our response to Reviewer rhsP below.

---

### Meta-Review · Area_Chair_yizp · 2021-08-10

**Recommendation:** Accept (Poster)
**Confidence:** 3

**Metareview:**

The reviewers didn't fully agree on the scores for the paper, but after the discussion with the authors, most of them felt that their points were adequately addressed in the rebuttal. I particularly encourage the authors to further clarify how this work is more than an incremental improvement over a baseline in [12] to address the remaining comments but I recommend this paper for acceptance at CoRL.

---

> ### Author Response · Authors · 2021-08-30
> **Response to Area Chair yizp**
>
> Dear Area Chair,
>
> We thank you for arranging the peer review of our manuscript and the four reviewers for their thorough reading of our work and insightful comments. We believe to have addressed all of the reviewers’ concerns in the new version.
>
> Best wishes,
>
> Authors

---

> > ### Comment · Area_Chair_yizp · 2021-08-30
> > **Thank you for your response**
> >
> > Thank you for your response! I'll wait for the reviewers to chime in before providing my own assessment.

---

### Decision · Program_Chairs · 2021-09-13

**Decision:**

Accept (Poster)

**Comment:**

The reviewers didn't fully agree on the scores for the paper, but after the discussion with the authors, most of them felt that their points were adequately addressed in the rebuttal. I particularly encourage the authors to further clarify how this work is more than an incremental improvement over a baseline in [12] to address the remaining comments but I recommend this paper for acceptance at CoRL.